# Comprehensive Engineering Frequency Domain Analysis and Vibration Suppression of Flexible Aircraft Based on Active Disturbance Rejection Controller

**DOI:** 10.3390/s22166207

**Published:** 2022-08-18

**Authors:** Litao Liu, Bingwei Tian

**Affiliations:** 1Institute for Disaster Management and Reconstruction, Sichuan University, Chengdu 610207, China; 2Pittsburgh Institute, Sichuan University, Chengdu 610207, China

**Keywords:** ADRC, vibration suppression, PID, Monte Carlo, frequency domain analysis, simulation

## Abstract

The crash of an aircraft with an almost vertical attitude in Wuzhou, Guangxi, China, on 21 March 2022, has caused a robust discussion in the civil aviation community. We propose an active disturbance rejection controller (ADRC) for suppressing aeroelastic vibrations of a flexible aircraft at the simulation level. The ADRC has a relatively simple structure and it has been proved in several fields to provide better control than the classical proportional-integral-derivative (PID) control theory and is easier to translate from theory to practice compared with other modern control theories. In this paper, the vibration model of the flexible aircraft was built, based on the first elastic vibration mode of the aircraft. In addition, the principle of ADRC is explained in detail, a second-order ADRC was designed to control the vibration model, and the system’s closed-loop frequency domain characteristics, tracking effect and sensitivity were comprehensively analyzed. The estimation error of the extended state observer (ESO) and the anti-disturbance effect were analyzed, while the robustness of the closed-loop system was verified using the Monte Carlo method, which was used for the first time in this field. Simulation results showed that the ADRC suppressed aircraft elastic vibration better than PID controllers and that the closed-loop system was robust in the face of dynamic parameters.

## 1. Introduction

In recent years, civil aviation companies have tended to adopt aircraft designs with larger aspect ratios to reduce their induced drag, achieve speed breakthroughs and reduce fuel consumption. However, this design sacrifices the aircraft’s stability and produces flight vibrations [1]. Compared to conventional designs, slender aircraft have larger wave resistance and are subject to increased reaction forces. Therefore, as civil aircraft are improved, more pronounced and sustained vibrations occur [2,3]. Corresponding to the fatigue theory of engineering mechanics, elastic vibrations in flexible aircraft can reduce their service life and may even damage the mechanical structure at certain specific frequencies, leading to more severe effects [3]. 

In recent years, scholars have been focusing on effectively suppressing the elastic vibration of aircraft to improve its performance. Chatter suppression techniques can be divided into two main categories: passive suppression and active suppression. Passive chatter suppression mainly seeks more stable operation at the expense of vehicle performance, such as increasing airframe weight and reducing flight distance [4]. Active suppression is mainly to effectively suppress the flutter phenomenon by optimizing the aircraft’s control system. Active flutter suppression (AFS) technology can be traced back to the research content of the famous project, active flexible wing (AFW), in the late 20th century [5]. Currently, the more mainstream AFS techniques include classical PID control, linear quadratic Gaussian (LQG) control, $H_{\infty }$ Control, sliding mode (SM) control, and fuzzy logic controller (FLC) [6,7,8,9,10,11,12,13,14,15,16,17,18,19,20,21,22,23,24,25,26], all of which play essential roles in modern control theory and have attracted the research interest of scholars. Figure 1 shows the development of some active flutter suppression techniques in chronological order. 

The invention of the PID control algorithm is primarily credited to Ziegler and Nichols, who introduced the concept of PID to the world in 1942 [6]. PID is the most widely used control algorithm and dominates the controller design subject. About 70 years later, Wang et al. developed a vibration model of the aircraft takeoff and landing process to facilitate the performance of the landing gear system through PID control [7]. After nearly a century of development, the functionality of PID has been over-exploited to the point that breakthroughs have been challenging. Many scholars have tried combining PID with different reference-seeking models in recent years to improve the control effect [8,9,10]. 

Gupta first proposed LQG control in 1980 and compared it with conventional controllers [11]. Ahmad, S., A. and Na, S., et al. have tried combining LQG with different feedback control strategies [12,13]. After 10 years of development, Franciszek combined the $H_{\infty }$ method with LQG to give a theoretical solution for nonlinear aeroelastic vibrations [14], the $H_{\infty }$ controller. Since its introduction in 1993, the control theory has been continuously discussed by scholars [15]. The works of Navya and Alexandr V. conducted a simulation analysis of the $H_{\infty }$ controller in the aerospace field [16,17]. The sliding model was first proposed in 1990 and applied to aerospace control at the beginning of the 21st century [18]. In the following 20 years, scholars have improved SMC strategies with different compensation methods [19,20,21]. In recent years, deep learning has brought epic changes in various fields. Han et al. proposed a controller combining SMC and reinforcement learning to achieve greater robustness and control accuracy [22]. FLC is also an essential player in modern control theory, and its performance has been widely discussed in the aerospace control field [23,24,25,26]. However, the researchers realized that there are several problems with the above algorithms, such as the PID algorithm seems too simple to adapt to industry application. The assumptions of modern control algorithms, such as LQG and SMC, are valid. However, the mathematical form is too complex and requires a high level of controller design, making it difficult to achieve a breakthrough from simulation to application.

The active disturbance rejection control (ADRC) algorithm, as an innovative product of PID theory, is expected by developers to achieve a breakthrough from theory to practical application of modern control algorithms. Jingqing Han formally proposed ADRC in 1998. The core of ADRC is the extended state observer (ESO), which estimates the system’s state and compensates for perturbations using the control method, turning the controlled object into a purely integral object [27]. Han pointed out the problems of PID algorithms and tried to use ADRC to solve the dilemma that PID has not made breakthroughs in development for decades [27,28,29]. Gao simplified Han’s work to obtain the linear active disturbance rejection control (LADRC) technique, which makes parameter tuning easier [30]. 

Regarding vibration suppression in aerospace, the control effectiveness of ADRC has been affirmed by researchers at the simulation level. Liu et al. combined an electromechanical actuator (EMA) with LADRC to propose a dynamic aircraft servo system [31]. Yang et al. used ADRC to suppress the elastic vibration of a high aspect ratio UAV fuselage [32]. Chen and Zhao proposed a Support Vector Machine (SVM) ADRC control strategy for UAVs to suppress the system chattering [33]. Wang et al. proposed an improved ADRC strategy to simulate and analyze hypersonic aerospace models [34]. Duan et al. modeled the helicopter load system to study the effect of load sway on the helicopter and confirmed the effectiveness of ADRC [35]. Wang et al. analyzed the limit cycle vibration suppression in hovering conditions. The results showed that ADRC has outstanding performance in suppressing limit cycle vibration and eliminating attitude control errors [36]. The work of Qiao and Zhong et al. brought innovation to ADRC control theory and applied it to the simulation analysis of aerospace control [37,38]. ADRC has been applied in some fields, such as financial [39] and industrial [40] fields, and some degree of progress has been made due to the application of ADRC. For example, in 2010, ADRC was first applied to a factory, reducing its energy consumption by 41% [40]. Therefore, ADRC algorithm’s role in aircraft controller design is also highly anticipated and has a high research value.

On 21 March 2022, the near-vertical crash of China Eastern Airlines plane MU5735 became a worldwide sensation. It was the deadliest air crash in China in 28 years. Across the globe, air crashes occur every year. Although air crashes do not occur as frequently as traffic accidents, the damage caused is enormous. After every air crash, there seems to be a warning that controllers need to be reformed to suit the needs of the airlines. In this study, we take a comprehensive engineering frequency domain analysis of the aeroelastic vibration model, mainly for flexible aircraft. In addition, we combined the developed aircraft elastic vibration model with the designed second-order active disturbance suppression controller to analyze the closed-loop control effect and the tracking performance and sensitivity of the closed-loop system. Finally, the Monte Carlo shooting method verified the system’s robustness.

## 2. Flexible Aircraft Aerodynamic Vibration Model 

Maki et al. created an aerodynamic vibration model of flexible aircraft in 1972 [41]. Furthermore, Zhong et al. built the aircraft model based on Maki’s theory in 2021 [38]. The general mathematical model of flexible aircraft pitch angle difference and the first elastic mode is given by [38,42]: (1)$$\left\{\begin{array}{l} \ddot{\theta }(t)=-b_{1}\dot{\theta }(t)+b_{2}(\theta (t)+\alpha )-b_{3}u(t)-b_{11}{\dot{q}}_{1}(t)-b_{21}q_{1}(t)+d_{1}(t),\\ {\ddot{q}}_{1}(t)=-2\xi \omega_{1}{\dot{q}}_{1}(t)-\omega_{1}^{2}q_{1}(t)+c_{1}\dot{\theta }(t)+c_{2}\theta (t)+c_{3}u(t)-d_{2}(t). \end{array}\right.$$

The two equations in Equation (1) represent the rigid body aerodynamic model channel of the aircraft pitch angle and the elastic vibration channel of the flexible aircraft, respectively. Figure 2 is a schematic diagram of the mathematical model in a flexible aircraft. Where $\theta \left(t\right)$ is the pitch angle difference, that is, the difference of the pitch angle at immediate posture (in time) and the initial posture, $u\left(t\right)$ is the control signal, $q_{1}\left(t\right)$ is the first generalized elastic mode of vibration, and it simulates the elastic vibration of the aircraft during flight. ξ and $\omega_{1}$ are the damping ratio and the natural frequency of the elastic mode, respectively, $c_{3}$ represents the control gain of the elastic mode $q_{1}\left(t\right)$, $c_{1}$, $c_{2}$, $b_{11}$ and $b_{21}$ are the coupling coefficients between the pitch angle and the first mode, α is the attack angle, $d_{1}\left(t\right)$ and $d_{2}\left(t\right)$ simulates the external disturbance of the system. 

The dynamic system’s Laplace-transfer matrix of zero initial condition is obtained. It is worth noting that the study in this section focuses only on the control object itself, so the external disturbance $d_{1}\left(t\right)$ and $d_{2}\left(t\right)$ are temporarily ignored in this section, and the angle of attack α = 0 is assumed.
(2)$$\left\{\begin{array}{l} \left(\frac{s^{2}+b_{1}s-b_{2}}{-b_{3}}\right)\Theta (s)+\left(\frac{b_{11}s+b_{21}}{-b_{3}}\right)Q(s)=U(s),\\ \left(\frac{-c_{1}s-c_{2}}{c_{3}}\right)\Theta (s)+\left(\frac{s^{2}+2\xi \omega_{1}s+\omega_{1}^{2}}{c_{3}}\right)Q(s)=U(s). \end{array}\right.$$

According to Gramer’s Law, the transfer function matrix is calculated:(3)$$\begin{array}{ll} G_{p} & =\left[\begin{array}{c} G_{1}\\ G_{2} \end{array}\right]=\left[\begin{array}{c} \frac{\Theta (s)}{U(s)}\\ \frac{Q(s)}{U(s)} \end{array}\right]\\ & =\left[\begin{array}{c} \frac{\left(s^{2}+2\xi \omega_{1}s+\omega_{1}^{2})(-b_{3})-(b_{11}s+b_{21})(c_{3}\right)}{\Delta (s)}\\ \frac{\left(c_{1}s+c_{2})(-b_{3})+(s^{2}+b_{1}s-b_{2})(c_{3}\right)}{\Delta (s)} \end{array}\right], \end{array}$$
Among them,
(4)$$\begin{array}{l} \Delta (s)=s^{4}+(2\xi \omega_{1}+b_{1})s^{3}+(\omega_{1}^{2}+2\xi \omega_{1}b_{1}-b_{2}+c_{1}b_{11})s^{2}+\\ (b_{1}\omega_{1}^{2}-2\xi \omega_{1}b_{2}+c_{2}b_{11}+c_{1}b_{21})s+c_{2}b_{21}-b_{2}\omega_{1}^{2}. \end{array}$$

$b_{1}=0.05$, $b_{2}=4$, $b_{11}=0.0005$, $b_{21}=0.05$, $c_{1}=0.05$, $c_{2}=30$, ξ=0.006, $\omega_{1}=30\;\mathrm{rad}/s$, $b_{3}=13$ and $c_{3}=330$ [38]. Bode plots of the transfer function $G_{1}$ and $G_{2}$ are shown in Figure 3.

For the transfer function $G_{1}$, the bandwidth $\left(w\right)$ is $w_{1}=1.28$ rad/s, the resonant peak $\left(Mr\right)$ and resonant frequency $\left(Rf\right)$ are $Mr_{1}=3.26$ dB, $Rf_{1}=0$ rad/s respectively, while for the transfer function $G_{2}$, $w_{2}$ is equal to 43.38 rad/s, $Mr_{2}$ is 30.60 dB, while $Rf_{2}$ is 30.00 rad/s (equals to $\omega_{1}$). In general, the system bandwidth is relatively large, so the system has a strong tracking capability for step signals. The resonant peaks and frequencies are relatively large, so the system is less stable in response to step signals.

## 3. Active Disturbance Rejection Controller Design

The ADRC control system consists of two main components, namelyu the extended state observer (ESO) and the proportional difference (PD) control method. In this section, we designed a second-order ADRC from these two parts, summarized how to simplify the ADRC parameter tuning, and analyzed the closed-loop system and its sensitivity.

### 3.1. Extended State Observer

Any dynamic system can be expressed by
(5)$$y^{\left(n\right)}+a_{1}y^{\left(n-1\right)}+\cdots +a_{n-1}y^{\left(1\right)}+a_{n}y=k(u^{\left(m\right)}+b_{1}u^{\left(m-1\right)}+\cdots +b_{m-1}u^{\left(1\right)}+b_{m}u)+w,$$
where $y^{\left(n\right)}$ refers to the nth order derivative of the control object output, $u^{\left(m\right)}$ refers to the mth order derivative of the control signal, w is unknown terms for all external disturbances, and the coefficient vectors $\vec{a}$ and $\vec{b}$ where any element ∈ℝ. The general expression of $\ddot{y}$ is obtained by
(6)$$\ddot{y}=f+b_{0}u.$$Among them, $b_{0}=b_{m}/a_{n-2}$, f contains all the terms in Equation (5) except $y^{\left(2\right)}$ and u. Therefore, f is expressed by a linear combination of other terms, that is
(7)$$f(y^{\left(n\right)},y^{\left(n-1\right)},\cdots ,u^{\left(m\right)},\cdots ,u^{\left(1\right)},w).$$Term f is defined as the generalized disturbance, including all perturbations inside and outside the system, which can be estimated by ESO to obtain $\hat{f}$. The equation below is a control law involved in ADRC:(8)$$u=\frac{-\hat{f}+u_{0}}{b_{0}}.$$
which $u_{0}$ will be discussed in PD controller design section. Rewrite Equation (6):(9)$$\ddot{y}=\left(f-\hat{f}\right)+u_{0},$$
where $\left(f-\hat{f}\right)$ is the estimation error, and Equation (6) can be rewritten as the following state-space equations:(10)$$\left\{\begin{array}{l} \dot{x}=Ax+Bu+E\dot{f},\\ y=Cx, \end{array}\right.$$Among them,
(11)$$\left\{\begin{array}{l} \dot{x}={\left[\begin{array}{ccc} {\dot{x}}_{1} & {\dot{x}}_{2} & {\dot{x}}_{3} \end{array}\right]}^{T}={\left[\begin{array}{ccc} \dot{y} & \ddot{y} & \dot{f} \end{array}\right]}^{T},\\ A=\left[\begin{array}{ccc} 0 & 1 & 0\\ 0 & 0 & 1\\ 0 & 0 & 0 \end{array}\right],B=\left[\begin{array}{c} 0\\ b_{0}\\ 0 \end{array}\right],E=\left[\begin{array}{c} 0\\ 0\\ 1 \end{array}\right],\\ C=\left[\begin{array}{ccc} 1 & 0 & 0 \end{array}\right]. \end{array}\right.$$

ESO is a stuff used to estimate $x_{1}$, $x_{2}$ and f. The estimated state-space equations are shown in the equation
(12)$$\left\{\begin{array}{l} \hat{\dot{x}}=A\hat{x}+Bu+E\hat{\dot{f}}+L\left(x_{1}-{\hat{x}}_{1}\right),\\ {\hat{x}}_{1}=C\hat{x}, \end{array}\right.$$
where $\hat{x}={\left[\begin{array}{ccc} \hat{y} & \hat{\dot{y}} & \hat{f} \end{array}\right]}^{T}$, $L\left(x_{1}-{\hat{x}}_{1}\right)$ is estimated compensation items, L is expressed by ${\left[\begin{array}{ccc} \alpha_{1} & \alpha_{2} & \alpha_{3} \end{array}\right]}^{T}$, is the gain vector of the ESO, and directly affects the performance of the observer. Rewriting Equation (12) yields:(13)$$\hat{\dot{x}}=\left(A-LC\right)\hat{x}+Bu+E\hat{\dot{f}}+Lx_{1}.$$

When the determinant of the matrix $\left(A-LC\right)$ is equal to zero, the system can converge.
(14)$$\det\left(\lambda I-\left(A-LC\right)\right)=\det\left(\left[\begin{array}{ccc} \lambda +\alpha_{1} & -1 & 0\\ \alpha_{2} & \lambda & -1\\ \alpha_{3} & 0 & \lambda \end{array}\right]\right).$$

The characteristic polynomial $p_{\left(n\right)}$ can be expressed as:(15)$$p_{\left(n\right)}=\lambda^{3}+\alpha_{1}\lambda^{2}+\alpha_{2}\lambda +\alpha_{3},$$
where $\alpha_{1},\alpha_{2},\alpha_{3}$ are the adjustable parameters. To simplify the work of parameter adjustment, the equation below has been proposed [30].
(16)$$p_{\left(n\right)}={\left(\lambda +\omega_{0}\right)}^{3},$$
where $\alpha_{1}=3\omega_{0}$, $\alpha_{2}=3\omega_{0}^{2}$, and $\alpha_{3}=\omega_{0}^{3}$.

### 3.2. PD Control Law

Assuming that the estimated value $\hat{f}$ has accurate estimation, then we obtain
(17)$$\ddot{y}=\left(f-\hat{f}\right)+u_{0}\approx u_{0}.$$Then, $u_{0}$ should be concerned, and the problem is solved by classical PD control law,
(18)$$u_{0}=k_{p}\left(r-{\hat{x}}_{1}\right)-k_{d}{\hat{x}}_{2}.$$By rewriting Equations (17) and (18), the transfer function of the PD control law can be obtained
(19)$$G_{pd}=\frac{k_{p}}{s^{2}+k_{d}s+k_{p}}.$$To simplify the PD law’s parameters, the equation below has been proposed [30],
(20)$$G_{pd}=\frac{k_{p}}{{\left(s+\omega_{c}\right)}^{2}},$$
where $k_{d}=2\omega_{c}$ and $k_{p}=\omega_{c}^{2}$, the final value theorem tells us that the pole occurs at $-\omega_{c}$, and if $\omega_{c}$ is greater than zero, the system will eventually stabilize [3].

### 3.3. Closed-loop Analysis

For the flexible aircraft vibration model in the previous section, it is suggested to use the second-order ADRC controller described above for its control. In order to simulate the natural data acquisition pattern during flight, it is assumed that there is a specific relationship equation between the first mode $q_{1}\left(t\right)$ and the pitch angle error $\theta \left(t\right)$ [38],
(21)$$\theta_{m}(t)=\theta (t)+0.2q_{1}(t),$$
where $\theta_{m}\left(t\right)$ is the state value obtained directly by the aircraft observer and 0.2 is the measurement factor.

Combining ESO with the vibration model and rewriting the state-space Equation (10) obtains
(22)$$\left\{\begin{array}{l} \dot{x}=\bar{A}x+\bar{B}u(t)+\bar{C}\dot{f},\\ y=\bar{D}x. \end{array}\right.$$

Among them,
(23)$$\left\{\begin{array}{l} x={\left[\begin{array}{ccc} \theta_{m}(t) & {\dot{\theta }}_{m}(t) & f \end{array}\right]}^{T},\\ f=-b_{1}{\dot{\theta }}_{m}(t)+b_{2}(\theta_{m}(t)+\alpha )+0.2{\ddot{q}}_{1}(t)\\ \left(-b_{11}+0.2b_{1}\right){\dot{q}}_{1}(t)+\left(-b_{21}-0.2b_{2}\right)q_{1}(t)+d_{1}(t),\\ \bar{A}=\left[\begin{array}{ccc} 0 & 1 & 0\\ 0 & 0 & 1\\ 0 & 0 & 0 \end{array}\right],\bar{B}={\left[\begin{array}{ccc} 0 & b_{0} & 0 \end{array}\right]}^{T},\\ \bar{C}={\left[\begin{array}{ccc} 0 & 0 & 1 \end{array}\right]}^{T},y=\theta_{m}(t),\\ \bar{D}={\left[\begin{array}{ccc} 1 & 0 & 0 \end{array}\right]}^{T}. \end{array}\right.$$

ESO estimates the state vector x and obtains the estimating equation:(24)$$\hat{\dot{x}}=\bar{A}\hat{x}+\bar{B}u(t)+\bar{C}\hat{\dot{f}}+L\left(\theta_{m}(t)-{\hat{\theta }}_{m}(t)\right),$$
where
(25)$$\left\{\begin{array}{l} \hat{x}={\left[\begin{array}{ccc} {\hat{\theta }}_{m}(t) & {\hat{\dot{\theta }}}_{m}(t) & \hat{f} \end{array}\right]}^{T},\\ L=\left[\begin{array}{ccc} 3\omega_{o} & 3\omega_{o}^{2} & \omega_{o}^{3} \end{array}\right]. \end{array}\right.$$

In order to obtain the control signal $u\left(t\right)$, it can be obtained by solving Equation (18), $u_{0}\left(t\right)$, which is then converted to $u\left(t\right)$ from Equation (8). Simplifying the progress obtains:(26)$$u\left(t\right)=\frac{-\hat{f}+2\omega_{c}\left(r-{\hat{\theta }}_{m}(t)\right)-\omega_{c}^{2}{\hat{\dot{\theta }}}_{m}(t)}{b_{0}}.$$

The aircraft model and the ADRC controller were successfully combined to form a feedback closed-loop (see Figure 4), where $G_{p}$ represents the controlled object and $G_{c}$ and F represents the active disturbance rejection controller (ADRC) [43].

The transfer function of the closed-loop system needed to be analyzed to predict the characteristics of the system. The transfer function of ADRC we designed can be given by [43]:(27)$$\left\{\begin{array}{l} G_{c}=\frac{k_{p}\left(s^{3}+\beta_{1}s^{2}+\beta_{2}s+\beta_{3}\right)}{b_{0}\left[s^{3}+\left(\beta_{1}+k_{d}\right)s^{2}+\left(\beta_{2}+s^{2}\beta_{1}+k_{p}\right)s\right]},\\ F=\frac{\left(k_{p}\beta_{1}+k_{d}\beta_{2}+\beta_{3}\right)s^{2}+\left(k_{p}\beta_{2}+k_{d}\beta_{3}\right)s+k_{p}\beta_{3}}{k_{p}\left(s^{3}+\beta_{1}s^{2}+\beta_{2}s+\beta_{3}\right)}. \end{array}\right.$$

The transfer function of the controlled object is given by Equation (3), and combined with Equation (27), the closed-loop transfer function $G_{cl}$ can be found:(28)$$G_{cl}=\left[\begin{array}{c} G_{cl1}\\ G_{cl2} \end{array}\right]=\left[\begin{array}{c} \frac{G_{c}G_{1}}{1+FG_{c}G_{1}}\\ \frac{0.2G_{c}G_{1}}{1+0.2FG_{c}G_{1}} \end{array}\right].$$

### 3.4. Parameter Tuning

By simplifying the parameters, there were finally three parameters left, namely, $b_{0}$, $\omega_{0}$ and $\omega_{c}$. The designed ADRC was simulated with the model for closed-loop feedback control, and the parameter values of the ADRC were trained using a unit-step setup signal. The parameter tuning of the model was limited by the range of values of some parameters in Table 1. Please refer to the following steps for the parameter tuning method.
As shown in Table 1, the six parameters of the elastic vibration model take random values within the intervals (simulation part), and each parameter has two boundary values. We know that if the system converges under a combination of 64 boundary values, any random set of parameters in the interval converge under the closed-loop system [38].Under the premise of step (a), select the mean value combination of the parameters of the vibration model, that is, $\omega_{1}=30,\;b_{2}=4,\;b_{3}=13,\;c_{3}=330,\;d_{1}=0\;and\;d_{2}=0$, continuously train the simulation effect of the closed loop, and analyze the simulation results. The maximum percentage overshoot (MPO-the percentage of maximum overshoot from the set value), setting time (TS-the earliest moment when the controlled object stabilizes in the interval [0.99,1.01]) and integral absolute error (IAE-the sum of the errors between the output value and the set value in the simulation time) are used as indicators to find the best simulation effect, and, then, the controller parameters are determined.

Through continuous training of the model, the controller parameters were finally determined, $\omega_{0}=7.8$ rad/s, $\omega_{c}=1.68$ rad/s and $b_{0}=-12.6,$ respectively. In real life, almost every field involving control can find the shadow of the PID controller. It is no exaggeration to say that among all controllers, PID is the most widely used controller at present. In order to show more clearly the advantages of ADRC compared to PID, a closed-loop containing a PID controller was considered. Considering the parameter tuning method mentioned above and the optimization parameters of the genetic algorithm, the robust characteristics of the PID controller were ensured [44], and the parameters of the designed controller were the proportional coefficient ($K_{P}=-0.7$), the integral coefficient ($K_{I}=-0.2$), the derivative coefficient ($K_{D}=-0.7$) and the filter coefficient (N = 1), respectively.

Figure 5 illustrates the tracking effect of ADRC and PID closed-loop results. The ADRC closed-loop control was far superior to the PID loop for pitch angle difference and control signal input. At the same time, the PID controller lost credibility in the face of the fourth-order instability object. Focusing on Figure 5a, for ADRC loop, the pitch angle error peaked at 1.2942 rad at 3.58 s with a maximum overshoot of 29.42%. The 1% setting-point was used as the upper and lower limits of the acceptable state. The setting time of the closed-loop was considered as the point where it first maintained the acceptable range, that is, $\theta_{m}\left(t\right)$ is in [0.99,1.01]. Therefore, the setting time of the ADRC loop was 8.65 s. Analyzing the control signal (Figure 5b), the minimum signal was minus 0.22 rad at 1.00 s, and the maximum signal was 0.43 rad at 3.34 s. Such a smooth control signal was appreciated. Overall, the designed ADRC controller achieved a good control effect by parameter adjustment. 

Moreover, the frequency domain of the closed-loop transfer function needed to be analyzed. Analyzing the rad solid-lines of Bode diagrams (Figure 6) of the closed-loop TFs Equation (28), for the transfer function $G_{cl1}$, the amplitude margin (Gm) was equal to 49.43 dB, and the phase margin (Pm) accounted for $60.34^{\circ }$, the cutoff frequency (Cf) was 30.00 Hz, while for $G_{cl2}$, the Gm=28.08 dB, the $Pm=-89.39^{\circ }$, and the Cf=19.04 Hz. The relatively high cutoff frequency of the closed-loop system reflected the more extraordinary ability of the system to cope with the dynamic response. The high amplitude margin indicated the higher inclusivity and stability of the closed-loop system.

### 3.5. Closed-Loop Sensitivity

Sensitivity is an essential indicator of the robustness of a closed-loop control system. For the closed-loop TFs (Equation (28)), the sensitivity function is given by [45]:(29)$$S=\frac{d\ln G_{cl}}{d\ln G_{p}}=\frac{dG_{cl}/G_{cl}}{dG_{p}/G_{p}}=\frac{G_{p}}{G_{cl}}\frac{dG_{cl}}{dG_{p}}.$$

Given that the parameters of the controller transfer function $G_{c}$ are fixed, while the parameter set of the control object $G_{p}$ is variable, bringing Equations (3) and (28) into Equation (29), we obtain, by rewriting:(30)$$S=\left[\begin{array}{c} S_{1}\\ S_{2} \end{array}\right]=\left[\begin{array}{c} \frac{1}{1+FG_{c}G_{1}}\\ \frac{1}{1+0.2FG_{c}G_{2}} \end{array}\right].$$

The Bode diagrams of $S_{1}$ and $S_{2}$ have been shown by the blue dotted-lines (see Figure 6). The Nyquist lines of $F\left(j\omega \right)G_{c}\left(j\omega \right)G_{1}\left(j\omega \right)$ and $0.2F\left(j\omega \right)G_{c}\left(j\omega \right)G_{2}\left(j\omega \right)$ are planned to be plotted in an imaginary coordinate system, and the point with the smallest distance from the plotted line to the coordinate (−1,0) is found, while the maximum value of sensitivity can be obtained. Its significance is elaborated in Wang’s book [46], where the maximum amplitude of the sensitivity $M_{s}$ is given by the following Equation:(31)$$M_{s}=\max\left|S\left(j\omega \right)\right|=1/r,$$
where r refers to the radius of inscribed circle with the center point (−1,0), which is tangent to the Nyquist graph line in the imaginary coordinate system (see Figure 7).

Figure 7 shows the results of the sensitivity analysis of the closed-loop system with two inscribed circles of radius $r_{1}$ and $r_{2}$, which were 0.62 and 0.57, respectively. Therefore, the maximum amplitude of the sensitivity was $Ms_{1}$ and $Ms_{2}$, which were 1.61 and 1.74, respectively. They were numerically equal to the resonance peaks analyzed from the Bode plot. The maximum amplitudes were less than 2, indicating that the system was robust in the face of parameter variations [46].

## 4. Simulation

In this section, the estimation error of ESO is first discussed, and then the parameters of the aircraft model are randomly selected according to the Gaussian distribution. Simulation results for each set of parameters are analyzed, and the system’s robustness is discussed. Figure 8 shows the mathematical model used for the simulation, implemented by the Simulink of Matlab.

### 4.1. ESO Estimation Error Analysis

A superiority of ADRC over the classic PID control method is its ability to provide an observational estimate of the state. The control effectiveness of ADRC for pitch angle vibration suppression is closely related to the accuracy of the ESO estimate.

As shown in Figure 9a–c, the maximum estimation error of $\theta_{m}\left(t\right)$ occurred at 1.19 s with an error of 0.02 rad, and the estimation of $\theta_{m}\left(t\right)$ by ESO was very accurate because it largely coincided with the true value. The maximum estimation error of ${\dot{\theta }}_{m}\left(t\right)$ occurred at 1.13 s with an estimation error of 0.64 rad/s. In general, the ESO’s estimation to ${\dot{\theta }}_{m}\left(t\right)$ was relatively accurate. For the total perturbation $f\left(t\right)$ estimation, the maximum estimation error of 14.78 rad/s^2^ occurred at 1.00 s. The effectiveness of ESO in estimating the total perturbation seemed less satisfactory, but such a result was acceptable because it included the total disturbance of the system, which had larger uncertainty. The estimation error of $f\left(t\right)$ stabilized in the range of [−1,1] after 2.77 s and, eventually, became stable as the system converged. Overall, the ESO estimates of the state variables largely overlapped with the system’s output response, demonstrating the superiority of ADRC.

### 4.2. Disturbance Rejection Test

The system’s immunity to interference should be mentioned, and it is recommended to consider the step disturbance terms $d_{1}\left(t\right)$ and $d_{2}\left(t\right)$ at 11 s, where $d_{1}\left(t\right)=0.07$ rad/s^2^ and $d_{2}\left(t\right)=-1.00$ m/s^2^ [38]. As shown in Figure 8d, the pitch angle maximum occurred at 18.56s at 1.0029 rad with a maximum overshoot percentage of 0.29%, which indicated that the system was always within the acceptable interval. At approximately 18s, the control signal $u\left(t\right)$ reached a maximum value of 0.294 rad and, finally, converged to a control signal of 0.293 rad. The subtle changes in pitch angle and control signal proved that the system had relatively good immunity to interference.

### 4.3. Robustness Analysis

In this section, we designed an innovative ADRC robustness testing method, the main idea of which is based on Monte Carlo statistical analysis using Gaussian distribution. Under the 3σ principle, it randomly draws the parameter values and composes 22,000 groups, guaranteeing a 99.73% confidence level in computer simulation. The random range of values for each parameter is given in Table 1. The median of each variable parameter was the mean value $\left(u\right)$ of the Gaussian distribution, and one-third of the difference between the mean and the upper or lower bound was used as the standard deviation $\left(\sigma \right)$. Figure 10 shows the Gaussian probability density function curves for the six parameters in Table 1.

Simulations performed for each set of parameters, and the outputs were analyzed. Three indices were planned to evaluate the system’s output: setting time (TS), maximum percentage overshoot (MPO), and integral absolute error (IAE). Of the 22,000 experiments perform, 364 had parameters outside the upper and lower bounds of random values (3σ principle). Therefore, only 21,636 sets of data were available.

Figure 11a,b show the simulation results for a total of 18,000 simulations, of which 17,706 sets of parameters satisfied the 3σ principal, and Figure 11c,d are the convergence curves saved at the 2000th and 8000th simulations, respectively, during the experimental process. From Figure 11a, there was a clear stratification of the setting time, which mainly occurred at 6.64 s, 8.92 s and 11.32 s. The second stratification occurred at 8.92 s, and the simulations with the setting time of less than 8.92 s accounted for 79.49% of the valid simulation. In addition, the amounts of simulation with a maximum overshoot percentage of less than 35% accounted for 78.59%, and the number when the integral absolute error was less than 1200 was 99.28%. Overall, the control system showed strong robustness in response to parameter changes.

Based on the experimental results, Table 2 counts three percentages under different simulation times. The first one was the proportion of valid simulations (parameters satisfying the 3σ principal). The second one was the ratio of the number of simulations with the setting time less than 8.92s to the number of valid simulations. As can be seen from Figure 11b, there was a strong linear relationship between the maximum percent overshoot and the integral absolute error, and we chose one of them as the third percentage in Table 2, which was the ratio of the number of simulations with the maximum percent overshoot less than 35% to the valid number of simulations. It can be seen from Table 2 that when the number of experiments reached more than 12,000 times, the changes of the three percentages were less than 0.1%. Therefore, for a 6-parameter Monte Carlo shooting test, the number of experiments was maintained at more than 12,000 times, and the number of experiments could be considered sufficient.

## 5. Discussion

The constant vibration frequency was the most significant deficiency of this study. In actual flight missions, the vibration frequency is variable, due to unknown external and internal disturbances. Therefore, it is recommended that deep learning algorithms, such as critical Recurrent Neural Networks (RNN) or Long Short-Term Memory (LSTM) neural networks, be considered in future studies to monitor and predict the system frequency changes at each moment, enabling ESO to achieve more accurate disturbance prediction. However, the combination of ADRC with deep learning algorithms can lose its usefulness to some extent.

## 6. Conclusions

This paper established a pitch angle vibration model of the flexible aircraft using Matlab/Simulink. The frequency domain characteristics of the system were analyzed comprehensively through a series of engineering analysis methods, while the design process of second-order ADRC was discussed in detail. Subsequently, an empirical summary of the ADRC parameter tuning process was proposed. In the presence of disturbances and uncertainties, the aircraft vibration model was controlled by two different robust control techniques, classical PID and ADRC techniques, respectively. Furthermore, the Monte Carlo shooting method was proposed to verify the robustness of the closed-loop control system. It was clarified that when the system parameters change in significant ranges, the designed controller showed a robust system within a 99.73% confidence interval by considering three indicators (MPO, TS, and IAE). In the test of referring to six dynamic parameters, more than 12,000 Monte Carlo simulations could reflect the statistical characteristics of the results. The structure of ADRC is also relatively simple compared with other modern control theories, and its control effect is better than the PID algorithm. The characteristics of robust control give ADRC the potential to evolve from simulation to practical engineering application.

## Figures and Tables

**Figure 1 sensors-22-06207-f001:**
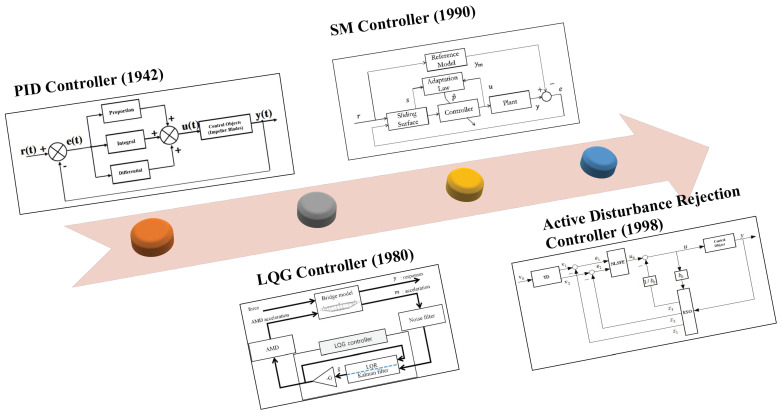
Timeline of the development of mainstream active flutter suppression controllers.

**Figure 2 sensors-22-06207-f002:**
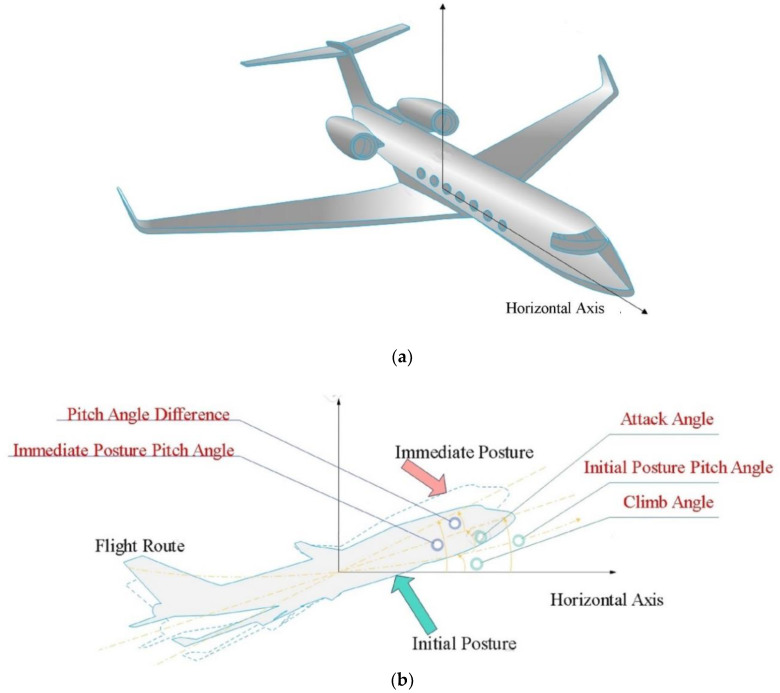
Schematic diagram of the elastic aircraft model. (**a**) the establishment of a flexible aircraft coordinate system; (**b**) schematic diagram of the mathematical model (Equation (1)). Among them, pitch angled difference represents physical quantity *θ*(*t*), and pitch angle is equal to climb angle plus attack angle (*α*).

**Figure 3 sensors-22-06207-f003:**
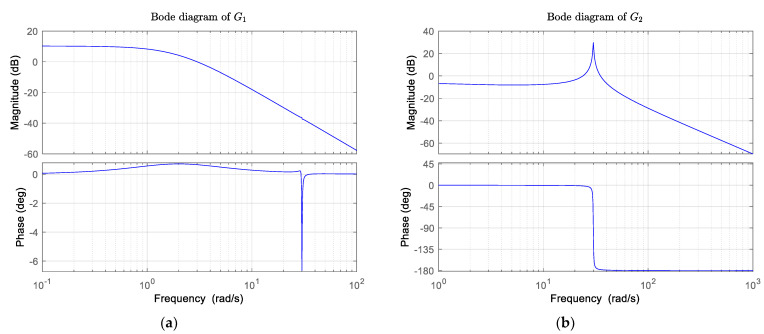
Bode plots of the transfer function. (**a**) the Bode plot of $G_{1}$; (**b**) the Bode plot of the $G_{2}$.

**Figure 4 sensors-22-06207-f004:**
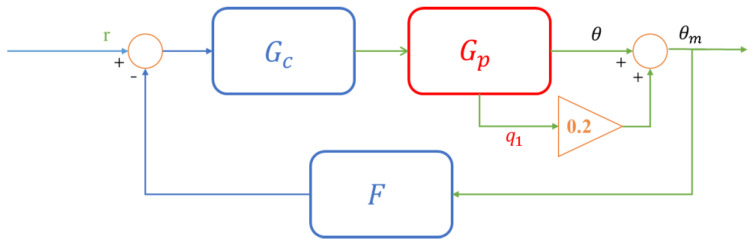
Closed-loop control system.

**Figure 5 sensors-22-06207-f005:**
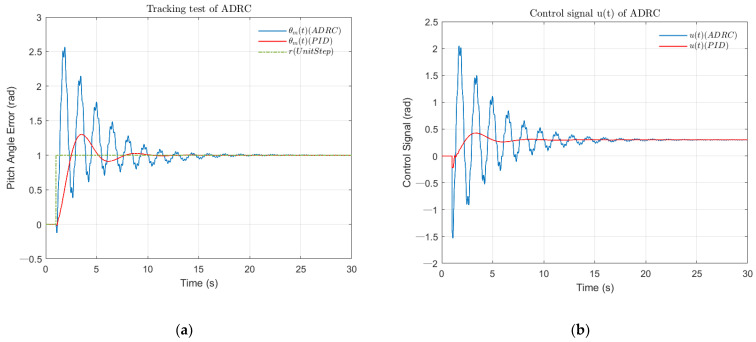
Comparison of the control effects of ADRC and PID controller (tracking test). (**a**) The simulation result of pitch angle of aircraft dynamic model under unit step signal; (**b**) The simulation result of control signal of aircraft dynamic model under unit step signal.

**Figure 6 sensors-22-06207-f006:**
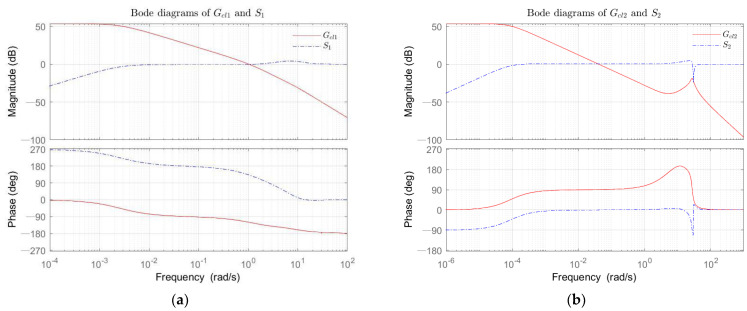
The Bode plots. (**a**) The Bode diagrams of closed-loop transfer function $G_{cl1}$ (rad solid-line), the sensitivity transfer function $S_{1}$ (blue dotted-line); (**b**) The Bode diagrams of closed-loop transfer function $G_{cl2}$ (rad solid-line), the sensitivity transfer function $S_{2}$ (blue dotted-line).

**Figure 7 sensors-22-06207-f007:**
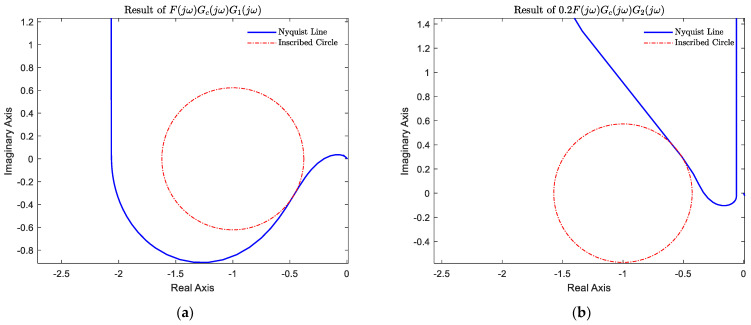
Results of sensitivity analysis. (**a**) Nyquist plot and sensitivity inscribed circles. $F\left(j\omega \right)G_{c}\left(j\omega \right)G_{1}\left(j\omega \right)$ (**b**) Nyquist plot and sensitivity inscribed circles.$\;0.2F\left(j\omega \right)G_{c}\left(j\omega \right)G_{2}\left(j\omega \right)$.

**Figure 8 sensors-22-06207-f008:**
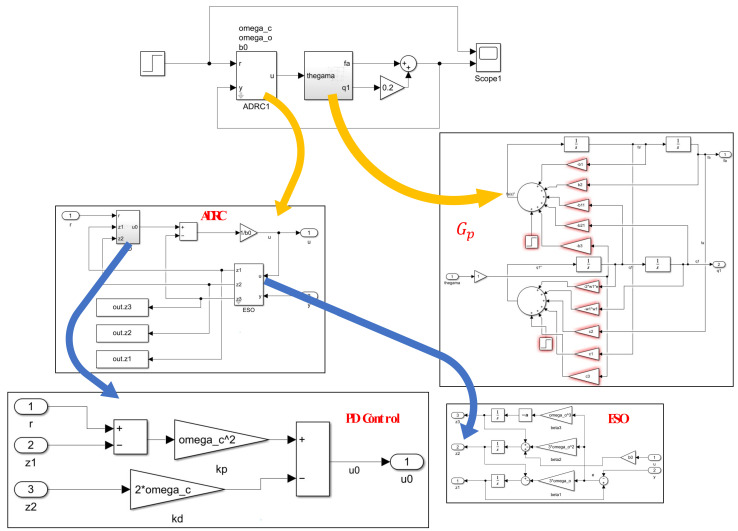
Simulation model building.

**Figure 9 sensors-22-06207-f009:**
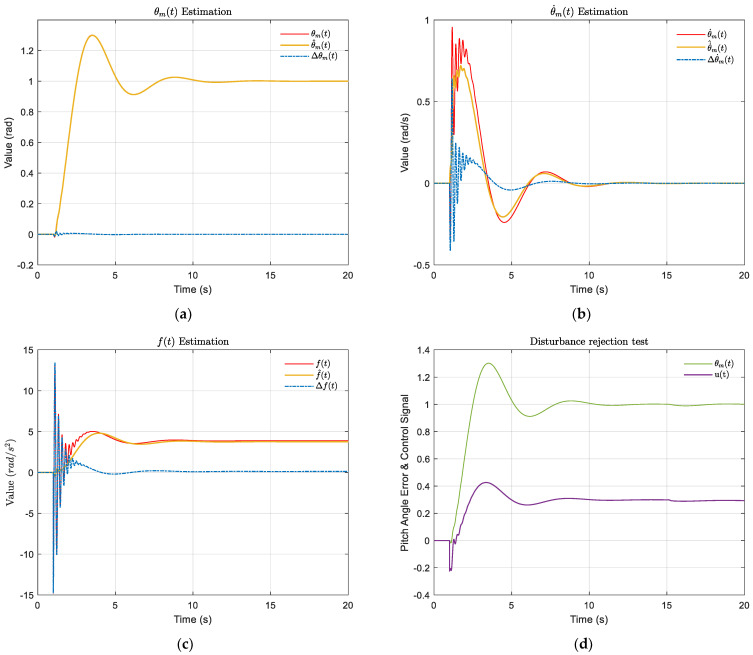
The estimated effects of ESO on states, and the disturbance rejection test. (**a**) The estimation of $\theta_{m}\left(t\right)$; (**b**) The estimation of ${\dot{\theta }}_{m}\left(t\right)$; (**c**) The estimation of $f\left(t\right)$ (the red solid-lines are real values; the yellow solid-lines are estimated values; the blue dotted-lines are the difference between real and estimated values.) (**d**) The disturbance rejection test result, the curve of pitch angle (green) and control signal (purple).

**Figure 10 sensors-22-06207-f010:**
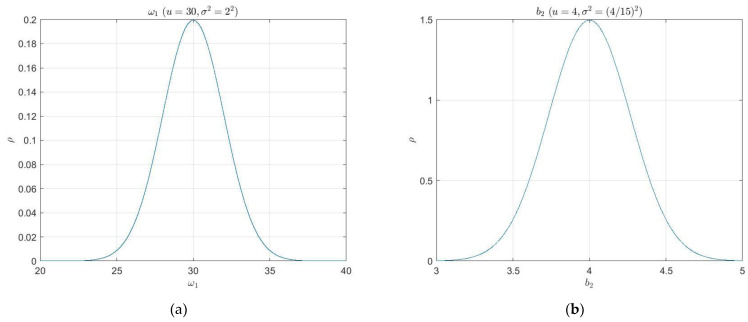
The Gaussian probability density function curves for the six variable parameters: (**a**) The natural frequency of elastic mode $\omega_{1}$; (**b**) The kinetic coefficient $b_{2}$; (**c**) The kinetic coefficient $b_{3}$; (**d**) The kinetic coefficient $c_{3}$; (**e**) The external disturbance of pitch angle $d_{1}$; (**f**) The external disturbance of elastic modes.

**Figure 11 sensors-22-06207-f011:**
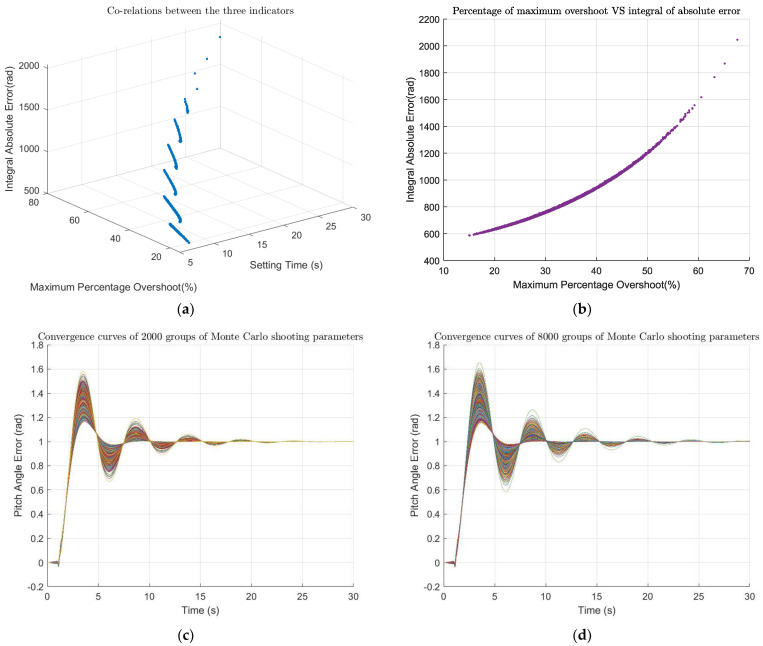
Monte Carlo shooting results. (**a**) The simulation results of the closed-loop system under the combination of the 17,706 valid parameters sets out of total 18,000 (the co-relations between all three indictors); (**b**) The simulation results and the correlation between maximum percentage overshoot and integral absolute error; (**c**) 2000 convergence curves drawn during simulation; (**d**) 8000 convergence curves drawn during simulation.

**Table 1 sensors-22-06207-t001:** Parameter variation range [38].

Term	Nomenclature	Value Range
$\omega_{1}$	The natural frequency of the elastic mode	[24, 36] rad/s
$b_{2}$	Kinetic coefficient	[3.2, 4.8]
$b_{3}$	Kinetic coefficient	[10.4, 15.6]
$c_{3}$	Kinetic coefficient	[264, 396]
$d_{1}$	External disturbance of pitch angle	[−0.07, 0.07] rad/s
$d_{2}$	External disturbance of elastic modes	[−1,1] rad/s^2^

**Table 2 sensors-22-06207-t002:** Statistical results under different simulation times.

Total Number of Monte Carlo Shooting	Valid Simulation (%)	Setting Time < 8.92 s (%)	Maximum Percentage Overshoot < 35% (%)
100	100.00	77.00	76.00
200	99.50	77.89	75.38
300	98.33	78.98	77.63
400	98.75	78.48	78.96
⋮			
1000	98.80	80.77	79.76
⋮			
5000	98.48	80.34	79.54
⋮			
10,000	98.34	79.18	78.24
⋮			
12,000	98.36	79.42	78.51
⋮			
15,000	98.43	79.44	78.53
⋮			
18,000	98.36	79.49	78.59
⋮			
22,000	98.35	79.41	78.51

The first column counts the total number of simulations, the second, third, and fourth columns count three percentages under different simulation times, valid simulation is the ratio of the amounts of valid simulations to the total simulation, and the third column counts the ratio of the number of simulations with the setting time less than 8.92s to the number of valid simulations. The fourth column counts the ratio of the number of simulations with the maximum percent overshoot less than 35% to the valid number of simulations. When the number of experiments is greater than 12,000, there are subtle changes in the three percentages (less than 0.1%).

## Data Availability

All data generated or analyzed during this study are included in this published article and its Appendix A.

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
