# Peer review of "Comprehensive Engineering Frequency Domain Analysis and Vibration Suppression of Flexible Aircraft Based on Active Disturbance Rejection Controller"

_sensors, 2022, doi:10.3390/s22166207_

Round 1

Reviewer 1 Report

The paper presents using an active disturbance rejection controller for aircraft vibration suppression. I have comments as follows.

1.      When you claim that “At present, the more mainstream active flutter suppression techniques include classical PID control, linear quadratic Gaussian (LQG) control, sliding mode (SM) control, fuzzy logic controller (FLC), etc.” Please provide references for those techniques. By the way, isn’t an h-infinity controller also popular?

2.      I am not familiar with a simple control model in Eq. (1) where it is claimed to represent the aircraft vibration model. Usually, when an aircraft is cruising or manoeuvring, it is subject to fluid-structure interaction from aeroelastic phenomena and from pilot control input etc. What I expect is the models detailed in chapter 8 of the ZAERO Theoretical Manual (https://www.zonatech.com/Documentation/ZAERO%209.3_THEO_Full_Electronic.pdf) onwards. Can you clarify how this model can be used with real aircraft?

3.      How can you obtain the PID controller coefficients on page 9 of the manuscript? Are those from optimisation? If not, how is performance validation in this paper justified?

4.      Why not compare to other techniques earlier mentioned in the literature review part e.g. linear quadratic Gaussian (LQG) control, sliding mode (SM) control, fuzzy logic controller (FLC), and h-infinity?

5.      For the Monte Carlo simulation, how can you ensure that 12000 points are enough for the reliability assessment? You may have to show the convergence curve by using (say) 5,000 points, 10,000 points, 12,000 points and 20,000 points. If the results from using 12,000 points and 20,000 points are slightly different, it is OK to use 12000 points. Also, it is quite common for robust control that not all the MCS points are valid, otherwise, the systems will be too conservative.

Please define all acronyms e.g. SVM-ADRC when they first appear in the abstract and the main text.

Author Response

We would like to express our sincere gratitude for your efforts in reviewing our manuscript. Please see the attachment for the response. Thank you! 

Reviewer 2 Report

The paper "Comprehensive Engineering Frequency Domain Analysis and Vibration Suppression of Aircraft Based on Active Disturbance Rejection Controller" must be improved.

In general the manuscript is badly written, leading the reader to do not understand the meaning of some sentences.

I suggest major revisions. Here below you can find some suggestions:

- The description provided in lines 130-134 is not sufficient; a figure may could help the reader to understand the system moedl.

- Eq 2, wn is not defined, maybe w1 in eq 1 becomes wn but still this reflect a low attention in writing.

- line 143 "Get the Bode plots" this is not scientific english.

- line 147 A resonant frequency equal to zero rad/s makes no sense if it is not related to a rigid mode. In this scenario such rigid mode should be carefully described.

- line 164 What is the reason for these assumptions?

- line 181-182 "is reasonable" what do you mean ?

- Eq20, how are the PD controller parameters tuned?

- line 238, "In real life,..." also other controllers are adopted in industrial applications

- Section 4 is badly written overall, I suggest improving its writing to help the reader understading.

Author Response

We would like to express our sincere gratitude for your efforts in reviewing our manuscript. Please see the attachment for the responses. Thank you! 

Round 2

Reviewer 1 Report

The paper can be accepted.